# Modulation of protein-DNA binding reveals mechanisms of spatiotemporal gene control in early *Drosophila* embryos

**Sahla Syed[1], Yifei Duan[1,2], Bomyi Lim[1]\***

[1]Department of Chemical and Biomolecular Engineering, University of Pennsylvania, Philadelphia, United States; [2]Master of Biotechnology Program, University of Pennsylvania, Philadelphia, United States

**Abstract** It is well known that enhancers regulate the spatiotemporal expression of their target genes by recruiting transcription factors (TFs) to the cognate binding sites in the region. However, the role of multiple binding sites for the same TFs and their specific spatial arrangement in determining the overall competency of the enhancer has yet to be fully understood. In this study, we utilized the MS2-MCP live imaging technique to quantitatively analyze the regulatory logic of the *snail* distal enhancer in early *Drosophila* embryos. Through systematic modulation of Dorsal and Twist binding motifs in this enhancer, we found that a mutation in any one of these binding sites causes a drastic reduction in transcriptional amplitude, resulting in a reduction in mRNA production of the target gene. We provide evidence of synergy, such that multiple binding sites with moderate affinities cooperatively recruit more TFs to drive stronger transcriptional activity than a single site. Moreover, a Hidden Markov-based stochastic model of transcription reveals that embryos with mutated binding sites have a higher probability of returning to the inactive promoter state. We propose that TF-DNA binding regulates spatial and temporal gene expression and drives robust pattern formation by modulating transcriptional kinetics and tuning bursting rates.

**\*For correspondence:**
bomyilim@seas.upenn.edu

**Competing interest:** The authors declare that no competing interests exist.

## Editor's evaluation

This valuable work explores how transcription factors regulate transcription through cooperative binding to enhancers. Through experiments and modeling, the authors show convincingly that the cooperativity of transcription factor binding regulates transcriptional bursting and the extent of the amount of time that the target promoter remains in an active state.

## Introduction

Development of a *Drosophila* embryo is a highly precise and coordinated process, occurring with little variability despite intrinsic and extrinsic noise and perturbations (*Arias and Hayward, 2006*; *Houchmandzadeh et al., 2002*). Proper levels of essential genes and correct positioning of expression patterns are regulated by short non-coding DNA sequences known as enhancers (*Banerji et al., 1981*). Enhancers tightly control their target gene expression both in space and time via transcription factor (TF) recruitment. Complex patterning and cell fates are established through TFs recognizing and binding to specific short DNA sequences within enhancers with varying degrees of affinity at different developmental stages (*Long et al., 2016*; *Reiter et al., 2017*; *Ramos and Barolo, 2013*). Concerted action of TFs with other transcriptional machinery has been found to reposition nucleosomes, initiate chromatin remodeling, recruit additional activating co-factors, and generate distinct transcriptional outputs (*Spitz and Furlong, 2012*). However, it remains to be understood how these

brief, yet frequent, interactions between TFs and regulatory DNAs facilitate efficient and specific transcription on the timescale of minutes. Although we know that TFs influence various facets of transcription such as timing or probability of activation, we have yet to determine their role in orchestrating an enhancer's transcriptional competency at a mechanistic level. For example, does the spatial arrangement of the binding sites influence transcriptional capability? How does each TF binding site shape transcriptional dynamics of individual nuclei and contribute to overall pattern formation?

Recently, it was shown that the genomic context of an enhancer provides an optimal environment for driving normal expression patterns and preventing misregulation upon induced perturbations. Mutating a single Giant (Gt) repressor binding site in the minimal *even-skipped* stripe 2 enhancer region caused misexpression of the target gene, whereas those effects were buffered in an extended enhancer containing more TF binding sites and expression levels were comparable to the wildtype (*López-Rivera et al., 2020*). However, the role of multiple TF binding sites with varying affinities within the enhancer in regulating transcription has yet to be established. Recent studies have explored the role of low-affinity binding sites in producing specific expression patterns and found that enhancers containing optimal TF motifs may lead to overexpression and result in developmental defects (*Farley et al., 2015*; *Ramos and Barolo, 2013*; *Tsai et al., 2017*). Previous work has shown that modulating the strength of a single TF binding site was sufficient to disrupt transcriptional activity, such that a mutation of an activator Dorsal (Dl) site in the *t48* enhancer delayed activation and almost completely abolished transcriptional activity, while optimization of the site to a consensus motif induced ectopic transcriptional activity with a broader gene expression domain (*Keller et al., 2020*). However, systematic removal of binding sites of varying affinities for another activator, Bicoid, seems to affect its target gene, *hunchback*, expression to a similar degree, indicating that each site has a nearly equal contribution to the overall expression pattern (*Eck et al., 2020*). Yet, since many studies have relied on fixed tissue experiments to derive the role of TFs in transcriptional regulation, the changes in real-time transcription kinetics that drive the observed misexpression are often overlooked. Dynamic interplay among TFs, cofactors, and DNA occurs on the order of seconds, a time resolution that cannot be resolved solely through RNAi and single molecule in situ hybridization experiments (*Mir et al., 2017*). Since TF binding events affect the expression of regulatory genes both spatially and temporally, incorporation of both live imaging techniques and predictive modeling is crucial to correlate transient TF-DNA binding to downstream transcriptional activity in single-cell resolution.

In this study, we investigated the effects of perturbing TF-DNA binding strength on the transcriptional dynamics of *snail* (*sna*) in early *Drosophila* embryos. *sna* is a well-characterized, key patterning gene that encodes a zinc finger protein and is responsible for the differentiation of the mesoderm (*Rembold et al., 2014*; *Leptin, 1991*; *Ip et al., 1992*). Sna represses the expression of genes responsible for neuroectoderm formation and establishes the mesoderm-neuroectoderm boundary (*Kosman et al., 1991*). Embryos lacking *sna* fail to undergo gastrulation, resulting in embryonic lethality (*Hemavathy et al., 2004*). Previous studies have demonstrated that *sna* expression is controlled by a proximal enhancer and a distal (shadow) enhancer located directly upstream and ~7kb upstream of the promoter, respectively (*Perry et al., 2010*). The distal enhancer is necessary for proper *sna* expression and the viability of the developing embryo, especially under genetic and environmental stresses (*Perry et al., 2010*; *Dunipace et al., 2011*). *sna* is a target gene of the Dl morphogen, and the nuclear gradient of maternally deposited Dl protein controls the sharp boundaries of *sna* expression, such that only nuclei with high concentrations of nuclear Dl express *sna* (*Figure 1B*; *Hong et al., 2008*). Through binding assays like EMSA and ChIP-seq, it was determined that the distal *sna* enhancer contains multiple, low-affinity binding sites for Dl, Twist (Twi), and the pioneer factor Zelda (Zld) (*Figure 1A*; *Zeitlinger et al., 2007*; *Ferraro et al., 2016*). Indeed, *sna* expression is completely abolished in embryos lacking Dl or Twi, and Zld null embryos show a delay in *sna* activation (*Dufourt et al., 2018*; *Liang et al., 2008*).

Here, we utilize a combination of quantitative live imaging and mathematical modeling to probe the underlying regulatory mechanisms that TFs employ to initiate transcription, regulate gene expression levels, and establish spatial boundaries. Using MS2-MS2 coat protein (MCP)-based live imaging, we visualized transcription dynamics driven by the wildtype minimal *sna* distal enhancer in the cases of with and without various TF binding site mutations. We find that mutating a single TF (Dl or Twi) binding site in the enhancer significantly reduces mRNA production of the target gene, mainly through lowering transcriptional amplitude by reducing RNA polymerase (Pol) II loading rate, without

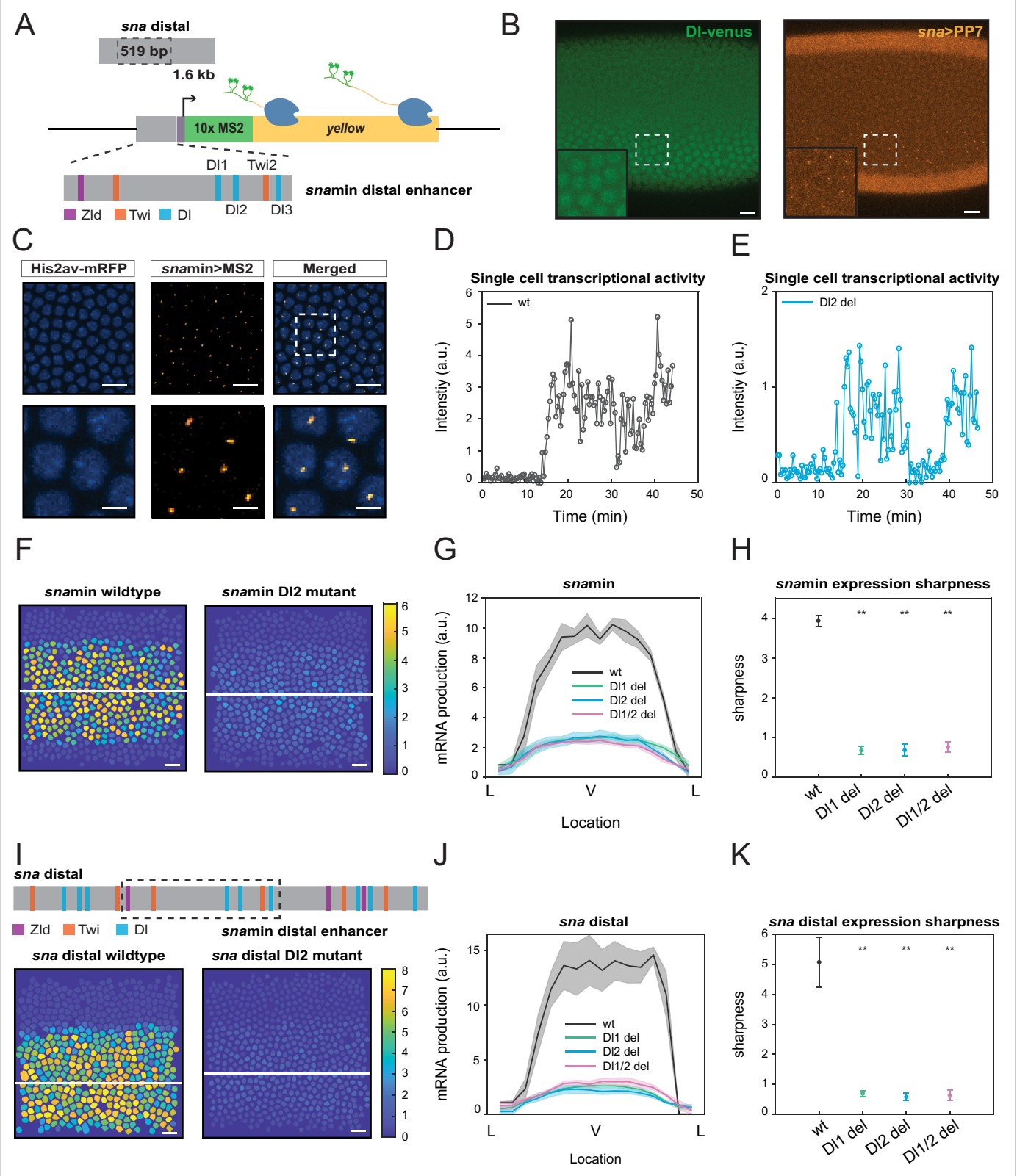

**Figure 1.** Single TF binding site mutation greatly diminishes mRNA production from the distal *sna* enhancer. (**A**) Schematic of the reporter construct containing the minimal *sna* distal enhancer, *sna* core promoter, MS2 stem loops, and the *yellow* reporter gene. The minimal enhancer contains binding sites for TFs Zld, Twi, and Dl. (**B**) Embryo expressing maternally deposited Dl-Venus protein (green) and *sna* distal>*PP7-yellow* reporter gene (orange). *sna* is expressed in the region with high nuclear Dl. The inset shows the region within the rectangle. Scale bar = 10 μm. (**C**) Snapshots of an embryo

*Figure 1 continued on next page*

*Figure 1 continued*

expressing minimal *sna* distal>*MS2-yellow*. The nuclei are marked with Histone-RFP (blue) and the MS2-*yellow* reporter gene is observed with MCP-GFP (orange). Each nucleus has one distinct fluorescent punctum, indicating nascent transcription. Scale bar = 10 µm. The bottom row are magnifications of the embryo within the rectangle. Scale bar = 5 µm. (**D–E**) Single nucleus transcriptional trajectories for a wildtype (**D**) and Dl2 mutant embryo (**E**). (**F**) Heatmap showing that mRNA production is higher in a wildtype embryo compared to a Dl2 mutant embryo. The white line indicates the ventral midline. Scale bar = 12 µm. (**G**) Average mRNA production of all nuclei in wildtype embryos and Dl1, Dl2, and Dl1/2 mutant embryos across the *sna* expression domain (Lateral-Ventral-Lateral). (**H**) Plot indicating the wildtype steepness of *sna* in expression is significantly higher than the mutants. Steepness was determined by calculating the maximum derivative of the mRNA output curves. (**I**) Top: Schematic showing the additional TF binding sites present in the full *sna* distal enhancer. Bottom: Heatmap showing higher mRNA production of *MS2-yellow* in a wildtype embryo containing the full *sna* distal enhancer compared to the embryo containing the full enhancer with Dl2 site mutations. Scale bar = 12 µm. (**J**) Average mRNA production of all nuclei in wildtype embryos and Dl1, Dl2, and Dl1/2 mutant embryos containing the full *sna* distal enhancer across the expression domain. (**K**) Plot indicating the wildtype steepness of *sna* distal expression is significantly higher than the mutants. Shaded error bars in (**G**) and (**J**) indicate the standard error of the mean (SEM). Error bars in (**H**) and (**K**) indicate SEM. A total of 1524 nuclei from three replicate wildtype embryos, 1672 nuclei from four replicate Dl1 mutant embryos, 2091 nuclei from four replicate Dl2 mutant embryos, and 1788 nuclei from three replicate Dl1/2 mutant embryos were analyzed. ** denote p<0.001 from the student's t-test.

The online version of this article includes the following figure supplement(s) for figure 1:

**Figure supplement 1.** Mutations affect mRNA production and expression sharpness, not width of expression domain.

significantly delaying the timing of initiation or affecting the probability of activation. Surprisingly, modulating the same TF binding site in the context of the full *sna* distal enhancer results in a similar reduction in expression levels, underscoring that additional TF binding sites in the full enhancer are not sufficient to rescue transcriptional activity. Using a thermodynamic equilibrium binding model, we show that the TF binding sites interact cooperatively to establish the correct spatial pattern of *sna*. Our model demonstrates that a combination of weak affinity and cooperative interactions among binding sites may be an evolutionary molecular mechanism to ensure the correct pattern and levels of target genes. Moreover, a two-state stochastic model of transcription indicates that TF binding site mutations affect transcriptional bursting, specifically by increasing the probability of the promoter switching out of the ON state, $k_{off}$, and reducing burst durations. Together, our data highlight the distinct mechanisms by which TF binding sites regulate transcriptional kinetics and spatial patterning during embryonic development.

## Results

### Single TF binding site mutation greatly diminishes the transcriptional capability of the distal *sna* enhancer

The 519 bp *sna* minimal enhancer is located within the full *sna* distal enhancer (1.6 kb) and has been shown to recapitulate proper *sna* expression (*Figure 1A*; *Ferraro et al., 2016*). We focus on the analysis of the minimal enhancer since it contains fewer TF binding sites than the full distal enhancer, which allows systematic perturbation in a sensitized background to gain a functional understanding of the role of TF binding site arrangements in gene regulation. The minimal enhancer contains binding sites for many TFs, including Dl, Twi, and Zld, and our study focuses on the sites with the strongest binding affinities, three Dl sites and one Twi site (*Figure 1A*). We note that all Dl and Twi sites still have relatively low binding affinities compared to the consensus motif, because *sna* is activated only in the domain with high nuclear Dl concentration (*Hong et al., 2008*). We systematically deleted one site at a time by introducing point mutations in each binding motif. The mutations were created by inducing

**Table 1.** Table of TF motifs and mutated sequences.
Table showing the sequences of the TF motifs, p-value, mutated sequences, and Patser score.

| Transcription factor | p-value | Patser score | Wildtype | Mutation |
|---|---|---|---|---|
| Dl1 | 1.43e-4 | 22.8 | AGGGATTTCCT | AGGGATCGCCT |
| Dl2 | 2.71e-05 | 19.8 | GGCGTTTTCCCA | GGCGATTGACCA |
| Dl3 | 8.52e-05 | 17.8 | TGGGAAATCGG | TGTTAAATCGG |
| Twi2 | 4.08e-05 | 7.8 | GTCCATGTGTTG | GTCCATGAATTG |

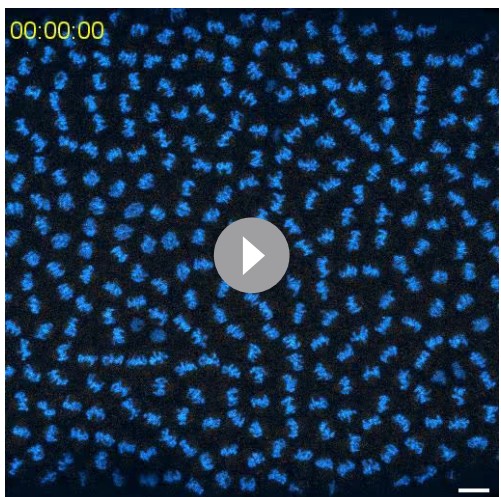

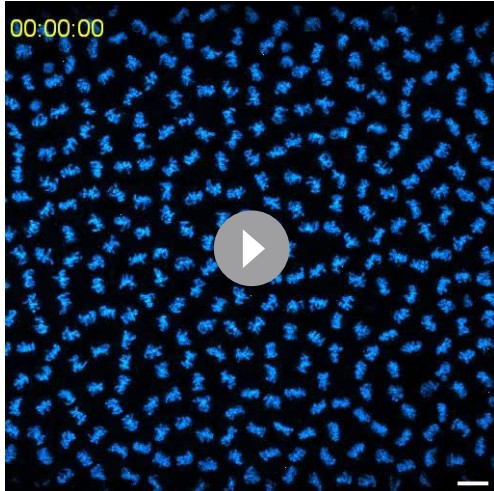

**Video 1.** Live imaging of wildtype embryo expressing snamin > MS2-yellow during NC14. MS2 signal is shown in yellow. Nuclei are marked with His2Av-mRFP. Histogram was adjusted for visualization purposes. Scale bar = 10 μm.

https://elifesciences.org/articles/85997/figures#video1

**Video 2.** Live imaging of Dl2 mutant embryo expressing snamin > MS2-yellow during NC14. MS2 signal is shown in yellow. Nuclei are marked with His2Av-mRFP. Scale bar = 10 μm.

https://elifesciences.org/articles/85997/figures#video2

nucleotide substitutions that cause the sequence to no longer be recognized as a TF motif match (*Table 1*).

We utilized the MS2-MCP live imaging technique to quantitatively analyze the effects of modulating TF binding sites within the minimal *sna* distal (sna*min*) enhancer. Specifically, 10x repeat sequences of MS2 are integrated into the 5′ UTR of the *yellow* reporter gene (*Figure 1A*). Upon transcription, each MS2 sequence forms a stem loop which is detected by maternally deposited MCP tagged with fluorescent proteins. To visualize nascent transcription dynamics in nuclear cycle (NC) 14, females carrying His2Av-mRFP (nucleus marker) and MCP-GFP were crossed with males carrying the desired MS2 construct, resulting in progeny expressing distinct fluorescent puncta in active transcription loci (*Figure 1C*, *Video 1*). *sna* distal enhancer drives gene expression in the endogenous *sna* domain, where the nuclear Dl level is the highest (*Figure 1B*). The fluorescence intensity trajectory for each nucleus is extracted for subsequent analysis and correlated to its instantaneous transcriptional activity (*Figure 1D and E*). Integration of the transcription trajectory over a given duration is proportional to the total number of mRNA molecules produced by a single nucleus (*Bothma et al., 2014*; *Garcia et al., 2013*). We do note that this does not provide an absolute number of mRNA molecules. However, through this estimation of mRNA production measured under the same laser setting, we can compare the relative mRNA production among constructs with different TF binding site mutations.

We found that a mutation of any single Dl or Twi binding site within the minimal enhancer resulted in a dramatic reduction in mRNA production (*Figure 1F*, *Videos 1 and 2*, *Figure 1—figure supplement 1A*). Nuclei in the center of the *sna* expression domain of a mutant embryo produce 65% less mRNA than those in the wildtype counterpart. The reduction occurs uniformly across the *sna* expression domain without significantly affecting the width (*Figure 1G*, *Figure 1—figure supplement 1B*). It is important to note that mutations of different TFs (Dl and Twi) and of binding sites with different affinities (Dl1 and Dl2) all result in a similar degree of decreased mRNA production. These results suggest that each site in the minimal enhancer is necessary to drive normal *sna* expression, perhaps due to the sensitized background in which the mutations were induced.

To further examine the significance of these core TF binding sites in the minimal enhancer, we investigated if the additional TF binding sites in the full distal enhancer would buffer against the drastic changes in mRNA production caused by the mutations. Despite several additional Dl and Twi sites flanking the minimal enhancer, we found that the same mutations on the same TF motifs caused a similar decrease in mRNA production (*Figure 1I and J*). Furthermore, we found that in both the minimal and full distal *sna* enhancers, the mutant embryos have shallower gradients and less sharp

boundaries, highlighting the importance of proper TF-DNA interactions in regulating the sharpness of the expression domain (*Figure 1H and K*, *Figure 1—figure supplement 1C*). Shallower expression of *sna* may lead to higher uncertainty in germ layer formation between mesoderm and neuroectoderm. It is interesting to note that the introduction of the mutations in the larger genomic context (i.e., full distal enhancer) did not lessen their effect on transcriptional activity and pattern formation. Here, it is evident that additional TF sites are not able to rescue normal *sna* transcriptional activity and that each site within the minimal enhancer region plays a critical role in ensuring robust expression.

## Mutations cause lower mRNA production, mainly due to reduced transcriptional amplitude and lower Pol II loading

After establishing the dramatic reduction in transcriptional activity, we delved into the underlying causes of the low mRNA production in both a single nucleus and across the entire *sna* expression domain. We hypothesized that the reduced mRNA production may occur through multiple different modes. The mutations may alter the time of transcriptional activation or reduce the transcriptional window, thereby effectively lowering the mRNA production compared to a wildtype embryo. The mutations may reduce the enhancer's ability to transcribe and effectively load Pol II, resulting in decreased instantaneous transcriptional amplitude. Or, the mutant embryos may have fewer transcriptionally active nuclei within the expression domain (*Figure 2A*). We found that the differences in the time for half the nuclei to begin transcription as well as the transcription initiation time per nucleus were not sufficient to explain the low mRNA output (*Figure 2B and C*, *Figure 2—figure supplement 1A*). Moreover, the total number of nuclei showing active transcription was comparable among wildtypes and binding site mutants (*Figure 4—figure supplement 1B*). Although the window of active transcription is slightly shortened in embryos containing the induced mutations (*Figure 2D*, *Figure 2—figure supplement 1B*), the main cause of the low mRNA output was the average transcriptional amplitude (*Figure 2E*, *Figure 2—figure supplement 1C*). Here, we observed a significant decrease in transcriptional intensity, leading us to conclude that the mutations mainly modulate transcription by lowering Pol II loading rate. At single-cell resolution, we find that the mutants with two deleted binding sites have a slightly bigger impact on transcriptional activity than those with one site removed, but the differences are small (*Figure 2C–E*, *Figure 2—figure supplement 1A–C*).

Since *sna* is a patterning gene and is responsible for the formation of the ventral furrow and presumptive mesoderm, we wanted to determine how the mutations spatially affect the aforementioned transcription parameters and the spatial boundaries of the expression pattern. The nuclei in the center of the *sna* expression domain are more substantially affected, exhibiting shorter window of transcription and lower average transcriptional amplitude, confirming the trend we observed for mRNA production (*Figure 2F–H*, *Figure 2—figure supplement 1D–F*). Our results agree with previous studies, in which the center nuclei within the *eve* stripe 2 domain had significantly longer periods of transcription and overall, higher rates of mRNA production (*Lammers et al., 2020*). Interestingly, we observed that the Dl1 mutation induced earlier transcriptional activation than wildtype (*Figure 2C and F*). Since the Dl1 and Dl2 binding sites are less than 10 bp apart, mutating the Dl1 site may create a more favorable steric conformation, allowing Dl to bind to the single site more efficiently. However, the earlier activation time is not sufficient to buffer against the severe reduction in transcriptional amplitude and causes the Dl1 mutant to exhibit similarly reduced mRNA production (*Figure 1G*).

Interestingly, in addition to modulating the average transcriptional intensity, the mutations in the full distal enhancer affect the time to activation and the transcription window as well. Mutant embryos have significantly more delayed transcriptional activation and a substantially shorter period of transcriptional activity (*Figure 2—figure supplement 2A–C*). As with the case with the minimal enhancer, we do not observe any significant spatial modulation of these parameters and the expression width remains unaffected (although the boundary is less sharp) (*Figure 2—figure supplement 2D–F*).

## Thermodynamic equilibrium binding model reveals synergistic interactions among TF binding sites

The mechanistic role that multiple TF binding sites with different affinities play in regulating enhancer activity and capability is still unclear. Is a single binding site sufficient to establish the correct pattern and expression levels? In that case, why does an enhancer contain multiple TF binding sites for the same TF? Our findings seem to indicate a nonadditive behavior between binding sites, such that

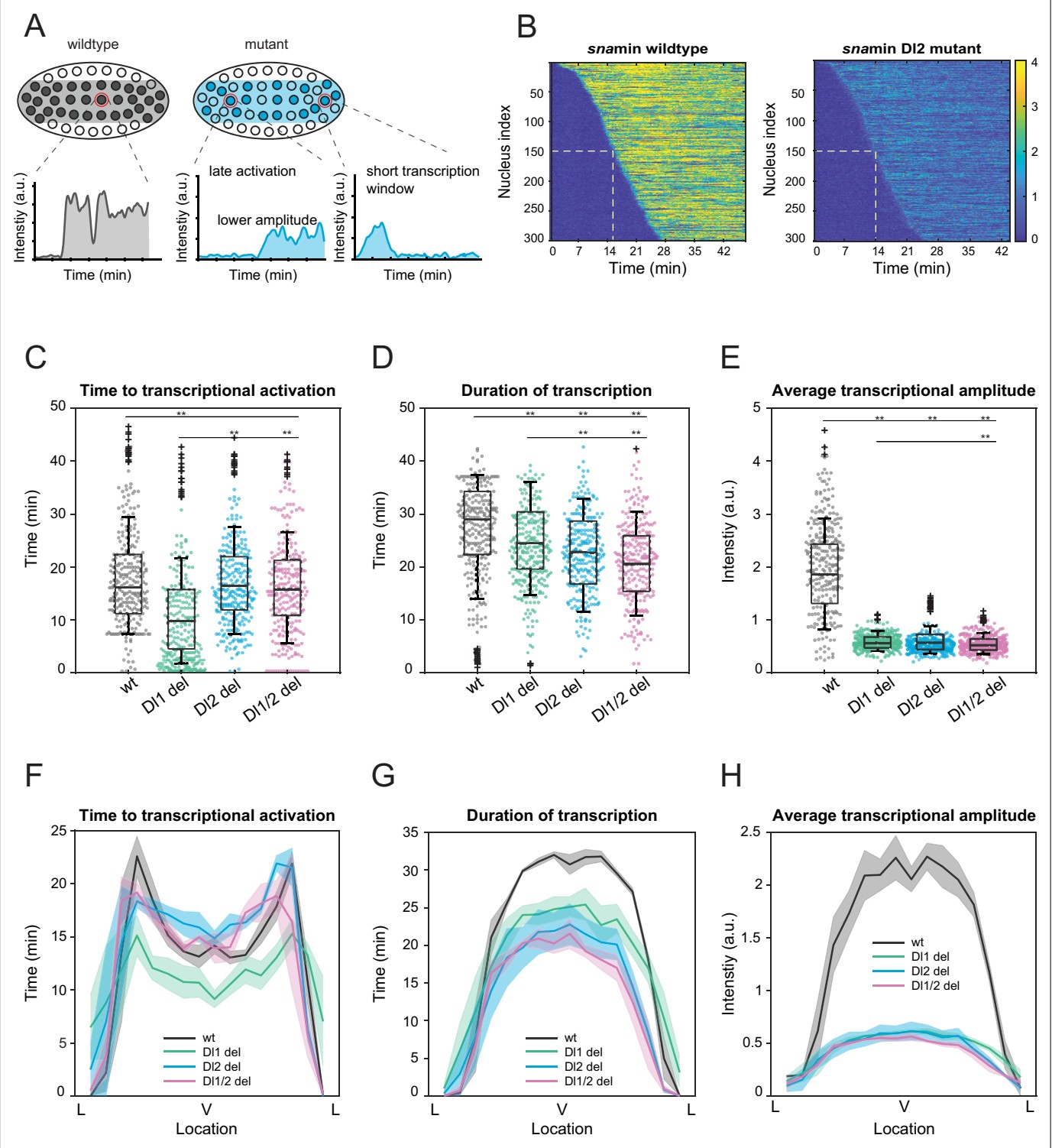

**Figure 2.** Mutations cause lower mRNA production, mainly due to lower transcriptional amplitude. (**A**) Schematic comparing wildtype and mutants. The mutant embryo may have fewer nuclei transcribing (shaded) and at a lower intensity. Each nucleus in mutant embryos may have late activation or a shorter transcription window, all of which may contribute to the observed low mRNA production. (**B**) Heatmap of transcription activation times for representative wildtype and Dl2 mutant embryos. The time at which half of the nuclei are activated is indicated by the dotted white line and there is no significant difference. (**C–E**) Boxplots showing (**C**) the time to transcriptional activation (**D**) the duration of active transcription, and (**E**) the transcriptional amplitude for all actively transcribing nuclei. Decreased transcriptional amplitude contributes the most to the low mRNA production in mutants. (**F–H**) (**F**) Average time to transcriptional activation, (**G**) average duration of transcription for all actively transcribing nuclei, and (**H**) average transcriptional

*Figure 2 continued on next page*

*Figure 2 continued*

amplitude for all nuclei across the *sna* expression domain (Lateral – Ventral – Lateral). Nuclei in the middle of the expression domain are affected more, but there is no significant change in the expression width. Shaded error bars in (**F–H**) indicate SEM. 250 individual data points are overlaid on the respective boxplots. A total of 1124 nuclei from three replicate wildtype embryos, 1011 nuclei from four replicate Dl1 mutant embryos, 1123 nuclei from four replicate Dl2 mutant embryos, and 943 nuclei from three replicate Dl1/2 mutant embryos were analyzed. Transcription of the *MS2-yellow* reporter gene is driven by the minimal *sna* distal enhancer. ** denote p<0.001 from the student's t-test.

The online version of this article includes the following figure supplement(s) for figure 2:

**Figure supplement 1.** Mutations cause lower mRNA production, mainly due to lower transcriptional amplitude.

**Figure supplement 2.** Mutations in the full *sna* distal enhancer decrease duration of transcription in addition to lowering transcriptional amplitude.

both a single and double mutation affect transcriptional dynamics to a similar degree. We utilized a thermodynamic Monod-Wyman-Changeux (MWC) model to examine the contributions of each Dl and Twi site to the overall competency of the enhancer (*Monod et al., 1965*). We assume that the microstates (unbound, bound with activator(s)) are in equilibrium and that the probability of each state can be correlated with its Boltzmann weight (*Eck et al., 2020*). As described in *Eck et al., 2020*;

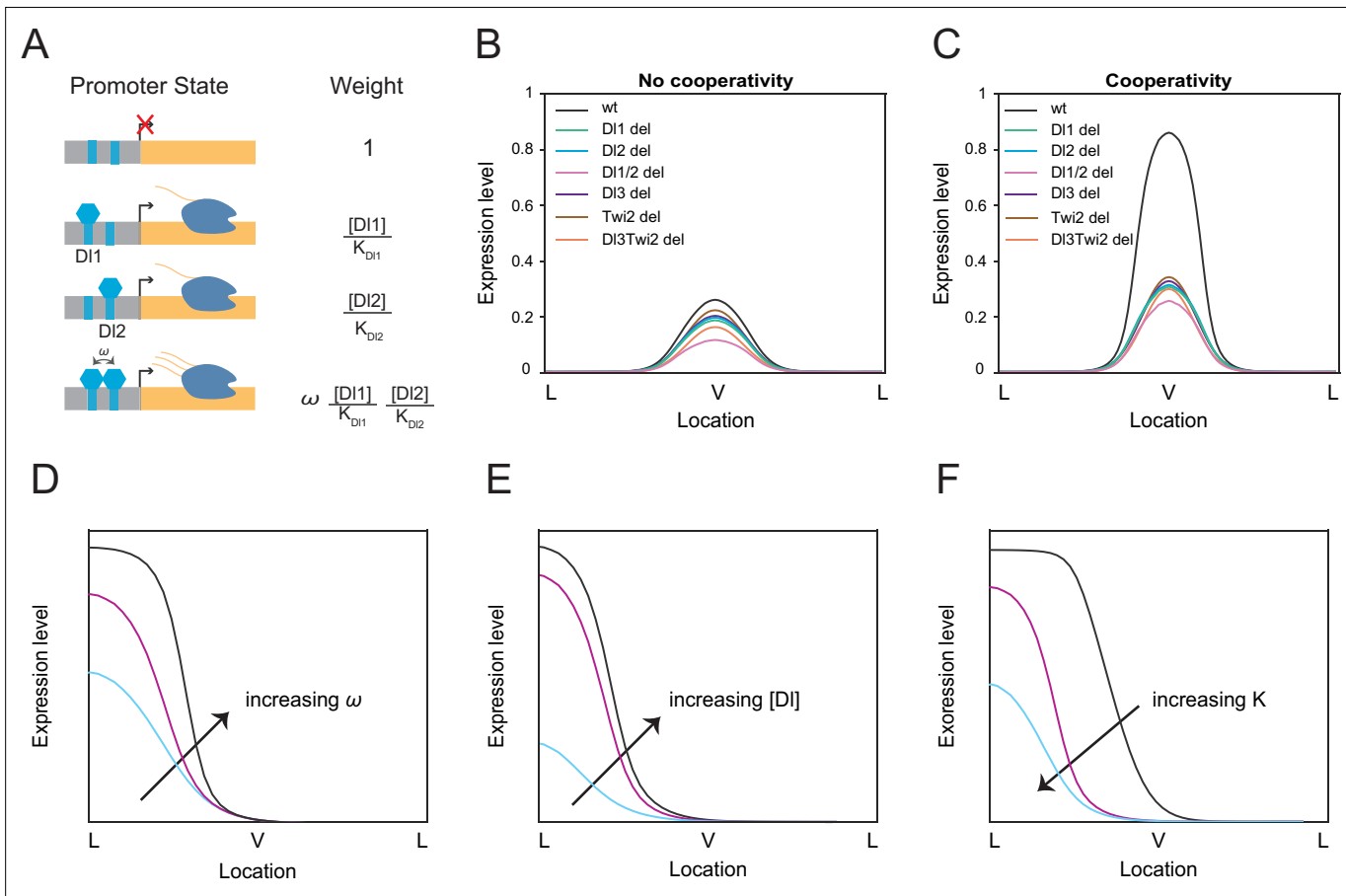

**Figure 3.** Thermodynamic equilibrium binding model reveals synergy among TF binding sites. (**A**) Promoter states and statistical weights for each microstate. A bound activator will yield transcription. Cooperativity term $\omega$ is included when more than one TF is bound, which will result in higher mRNA production. Dissociation constants are given by $K_i$, which is correlated with the binding affinity of each site. (**B–C**) mRNA production curves generated by assuming (**B**) no cooperativity and (**C**) cooperativity among TF binding sites. Modeling results support experimental data with the cooperativity term included. (**D–F**) Summary models noting expression level changes as cooperativity increases (**D**), as Dl concentration is increased in low binding affinity enhancers (**E**), and as binding affinity is increased (**F**).

The online version of this article includes the following figure supplement(s) for figure 3:

**Figure supplement 1.** Sensitivity analysis on thermodynamic model highlights the importance of cooperativity among all TFs.

**Figure supplement 2.** Modulation of binding affinity and Dl concentration.

*Kanodia et al., 2012*, the Boltzmann weights can be calculated in terms of activator concentration $C$, dissociation constant $K$, and a cooperativity term $\omega$ (*Figure 3A*). The probability of transcription initiation is represented by the probability of an activator (i.e., Dl) binding to its cognate site divided by the sum of all possible states, which can be written as:

$$p_{bound} = \frac{\dfrac{C_{Dl1}}{K_{Dl1}} + \dfrac{C_{Dl2}}{K_{Dl2}} + \omega\dfrac{C_{Dl1}C_{Dl2}}{K_{Dl1}K_{Dl2}}}{1 + \dfrac{C_{Dl1}}{K_{Dl1}} + \dfrac{C_{Dl2}}{K_{Dl2}} + \omega\dfrac{C_{Dl1}C_{Dl2}}{K_{Dl1}K_{Dl2}}}$$

**Table 3.** Table of promoter occupancies with and without cooperativity.
Table showing the promoter occupancies in the cases of cooperativity and no cooperativity for the three cooperativity terms that were greater than 1 in the optimized model. Cooperativity increases promoter occupancy by 90%.

| Promoter state | Promoter occupancy with cooperativity | Promoter occupancy with no cooperativity |
|---|---|---|
| $\omega_1$ | 2.4378 | 0.0870 |
| $\omega_6$ | 2.1913 | 0.0397 |
| $\omega_{11}$ | 54.8073 | 3.45e-04 |

This equation can be expanded to include all the combinations of Dl and Twi interactions (See Methods, *Equation 1*). The equilibrium binding constants were correlated to their binding affinity scores according to the Patser algorithm (*Table 1*) using the Dl position weight matrix (PWM) (*Hertz and Stormo, 1999*; *Alamos et al., 2023*; see Materials and methods).

We tested our model in the case of completely independent binding, or no cooperativity. In this condition, the model failed to predict the experimental data and we found that the wildtype has only slightly higher expression levels compared to the mutants (*Figure 3B*). When we modified the model to include the cooperativity terms among all the TFs, we were able to qualitatively recapitulate the experimental mRNA production curve as well as determine which TFs work synergistically to help recruit more TFs to the enhancer region (*Figure 3C*). Promoter occupancy is significantly higher when TFs interact cooperatively, such that removing cooperativity reduces promoter occupancy by more than 90% (Table 3). We do note that the model predicts that the Dl1/2 double mutant exhibits lower mRNA output than the other single binding site mutants, which is not as prominent in experimental results (*Figure 2C–E*). We believe this may be due to some degree of buffering from additional TF binding sites present in the *sna* enhancer that cannot be captured in our four binding site model. We observe the same trend of a lower mRNA output for the double binding site deletion regardless of the changes in binding site affinity (*Figure 3—figure supplement 2A–B*).

**Table 2.** Table of cooperativity values.
Table showing the cooperativity values for the optimized model and the cooperativity values when all binding sites are weakened or strengthened. Cooperativity values are much higher when the strength of the binding sites are weakened.

| Cooperativity term | Transcription factor | Optimized value | Weak binding value | Strong binding value |
|---|---|---|---|---|
| $\omega_1$ | Dl1, Dl2 | 28.00 | 69.90 | 2.03 |
| $\omega_2$ | Dl2, Twi2 | 1.00 | 1.00 | 1.00 |
| $\omega_3$ | Dl1, Twi2 | 1.00 | 1.00 | 1.00 |
| $\omega_4$ | Dl1, Dl3 | 1.00 | 1.00 | 1.00 |
| $\omega_5$ | Dl2, Dl3 | 1.01 | 1.00 | 1.00 |
| $\omega_6$ | Dl3, Twi2 | 55.30 | 139.00 | 6.79 |
| $\omega_7$ | Dl1, Dl2, Twi2 | 1.07 | 1.05 | 1.03 |
| $\omega_8$ | Dl1, Dl2, Dl3 | 1.03 | 1.02 | 1.01 |
| $\omega_9$ | Dl2, Twi2, Dl3 | 1.13 | 61.10 | 1.03 |
| $\omega_{10}$ | Dl1, Twi2, Dl3 | 1.03 | 1.03 | 1.02 |
| $\omega_{11}$ | Dl1, Dl2, Twi2, Dl3 | 1.59e5 | 5.00e5 | 3.10e4 |

We found that Dl1/Dl2 ($\omega_1$), Dl3/Twi2 ($\omega_6$), and Dl1/Dl2/Dl3/Twi2 ($\omega_{11}$) cooperativities were the key parameters with a value above 1 (i.e. positive cooperativity) to reach a stable solution, confirming the role of cooperativity among weak binding sites as a mechanism for precise gene control (*Table 2*; *Table 3*). We performed sensitivity analysis on these three cooperativity terms to test the robustness of the model. Transcriptional output was not significantly affected upon modulating a moderate cooperativity $\omega_1$ and $\omega_6$ values by an order of magnitude (*Figure 3—figure supplement 1A and B*). However, the model was more sensitive to changes in the highest cooperativity term $\omega_{11}$, which represents the interactions among all TFs (*Table 2*). Decreasing the cooperativity value of $\omega_{11}$ results in a smaller difference between wildtype and mutant transcriptional activities, while increasing cooperativity results in ectopic expression in a wider expression domain (*Figure 3—figure supplement 1C*). This underscores the importance of the concerted interaction among all the TFs in regulating and maintaining proper gene expression pattern and levels (*Figure 3D*).

We sought to obtain comprehensive insights into the mechanism of binding site interactions by systematically modulating model parameters. The *sna* enhancer contains all relatively weak Dl binding sites and activates expression only in the presence of high Dl concentration. To dissect the role of weak versus strong sites in enhancers regulating expression under different morphogen concentrations, we varied nuclear Dl levels. Decreasing Dl concentrations in the presence of low affinity binding sites results in a very low transcriptional activity, perhaps because the TF-DNA interactions are limited both by low TF concentration and low TF-DNA affinity (*Figure 3—figure supplement 2C*). In order to obtain high transcriptional activity under a low Dl concentration background, the affinity of the TF binding sites must be increased by a factor of 2 (*Figure 3—figure supplement 2D*). This suggests that enhancers of the genes activated by lower concentration of Dl should contain strong binding sites to maintain transcriptional activity at normal levels (*Figure 3E*; *Hong et al., 2008*). It will be of further interest to quantitate how the induced mutations affect *sna* activity in the presence of reduced maternally deposited Dl levels.

Next, we utilized our model to test the effects of strengthening or weakening one binding site on transcriptional output. Further weakening Dl1 binding affinity renders it an almost non-affinity site and results in overall reduced mRNA production, while strengthening Dl1 affinity leads to an ectopically wider expression (*Figure 3—figure supplement 2E–F*). Relying on a single strong site may yield 'too high' and ectopic expression, and its removal can drastically disrupt expression, a result that was also observed upon removal of a single Dl site in the *t48* enhancer (*Figure 3F*; *Keller et al., 2020*).

Taken together, our data provides evidence that TF binding sites should coordinate with one another to some degree to recreate the correct pattern and levels of *sna*. We propose that cooperativity allows TF binding sites with moderate or weak affinities to recruit more TFs to the enhancer, generate sharp transcriptional responses, and drive strong and robust expression in the narrow *sna* expression domain (see Discussion).

## A two-state model of transcription reveals differences in k<sub>off</sub> rates and burst duration

Previous studies have shown that transcription occurs discontinuously in distinct, stochastic bursts of activity punctuated by quiescence (*Corrigan et al., 2016*; *Senecal et al., 2014*; *Donovan et al., 2019*). Bursting has been proposed as an evolutionary mechanism for driving heterogeneity in gene expression, giving rise to cell-to-cell variability and overall diversity (*Rodriguez and Larson, 2020*). Bursting parameters, such as burst frequency, burst duration, and promoter switching rates provide a glimpse into the underlying mechanisms of dynamic transcription regulation, such as kinetic rates and promoter states. We find that the wildtype and mutant embryos show comparable numbers of actively transcribing nuclei at the beginning of the NC14. However, at later times, most wildtype nuclei in the *sna* expression domain are active in a given frame while nuclei from mutant embryos exhibit stochastic activity (*Figure 4A*, *Videos 3 and 4*). Indeed, by quantifying the number of nuclei transcribing at every time point, we find that the embryos with mutations have far fewer active nuclei in each frame compared to the wildtype, despite their cumulative number of active nuclei being comparable (*Figure 4—figure supplement 1A and B*). We use stochastic modeling of transcription to investigate if TF binding site mutations cause changes in transcriptional bursting characteristics of each nucleus. The two-state model, in which a promoter can switch between an active (ON) and inactive (OFF) state, has been widely implemented to gain

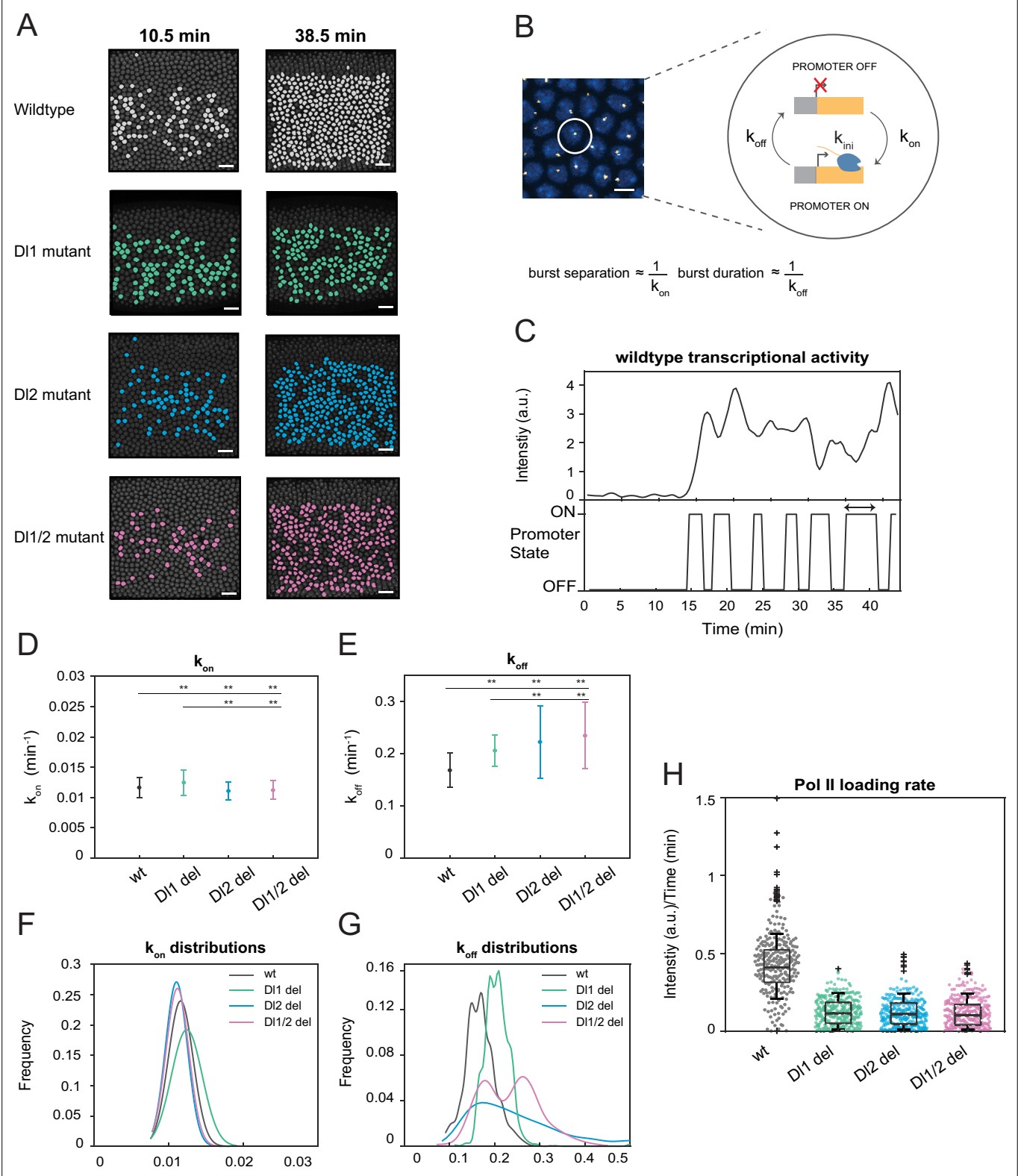

**Figure 4.** A two-state model reveals differences in $k_{off}$ rates and burst duration. (**A**) Actively transcribing nuclei are false-colored for early and late NC14. Mutant embryos show more sporadic transcriptional activity in a given frame. Scale bar = 15 µm. (**B**) Schematic depicting a single nucleus that can be in an OFF or ON state and switches between the two with rates $k_{on}$ and $k_{off}$. The rates of $k_{on}$ and $k_{off}$ can be correlated to burst separation and burst duration, respectively. Scale bar = 5 µm. (**C**) Representative transcriptional trajectory of wildtype with the inferred promoter states derived from the

*Figure 4 continued on next page*

*Figure 4 continued*

Hidden Markov model (HMM). Arrow indicates burst duration. (**D–E**) Plots showing the rates of (**D**) $k_{on}$ and (**E**) $k_{off}$. $k_{on}$ is not significantly affected, whereas $k_{off}$ rates are higher in the mutant embryos. (**F–G**) Probability distributions of (**F**) $k_{on}$ rates and (**G**) $k_{off}$ rates for wildtype, Dl1, Dl2, and Dl1/2 mutants. The distribution of $k_{on}$ rates follow a tight normal distribution while the $k_{off}$ distributions vary widely. (**H**) Boxplot showing the Pol II loading rates of all actively transcribing nuclei. The wildtype embryos have a significantly higher rate of Pol II loading than the mutant embryos. Error bars in (**D–E**) indicate standard deviation (SD). A total of 1124 nuclei from three replicate wildtype embryos, 1011 nuclei from four replicate Dl1 mutant embryos, 1123 nuclei from four replicate Dl2 mutant embryos, and 943 nuclei from three replicate Dl1/2 mutant embryos were analyzed. Transcription of the MS2-yellow reporter gene is driven by the minimal *sna* distal enhancer. ** denote p<0.001 from the student's t-test.

The online version of this article includes the following figure supplement(s) for figure 4:

**Figure supplement 1.** A two-state model reveals differences in $k_{off}$ rates and burst duration.

**Figure supplement 2.** Mutations in the full *sna* distal enhancer increase $k_{off}$ rates leading to reduced burst duration.

a functional understanding of bursting control on transcription (*Figure 4B*; *Bothma et al., 2014*; *Corrigan et al., 2016*).

Hidden Markov models (HMMs) are utilized to reveal hidden states not directly observable based on a sequence of observed events (*Bronson et al., 2009*). They are extensively used to recover rates of promoter switching and other bursting parameters from transcriptional trajectories (*Bothma et al., 2014*; *Lammers et al., 2020*; *Hoppe et al., 2020*). In this study, we utilize an HMM to infer the promoter state based on the observed fluorescence intensity curves (*Figure 4C*). Based on previous studies, burst separation and burst duration can be correlated to bursting parameters, $k_{on}$ and $k_{off}$ (*Hoppe et al., 2020*; *Zoller et al., 2018*). Using our modeling approach, we extracted the kinetic rates of the promoter returning to an active state from an inactive state ($k_{on}$) and vice versa ($k_{off}$). Previously, we demonstrated that the mutations affect mRNA production through transcriptional amplitude (*Figure 2E*). This could be because either the mutations hinder Pol II loading rate or reduce the time the promoter is in the ON state. Our results reveal that $k_{on}$ is only slightly affected by the induced mutations, while $k_{off}$ is significantly increased in mutants (*Figure 4D and E*, *Figure 4—figure supplement 1C and D*). Furthermore, all of the perturbations do not change the normal distribution trend of $k_{on}$ rates (*Figure 4F*, *Figure 4—figure supplement 1F*). However, we observe heterogeneity and high variability in the distribution of $k_{off}$ rates as well as shorter burst durations in the mutated embryos, explaining the bursty and noisy transcriptional activity we observed (*Figure 4G*, *Figure 4—figure supplement 1E and G*). This leads us to conclude that the mutations affect the ability of the promoter to remain in the active state, causing it to become more unstable and more likely to revert to the inactive state, supporting the observation of lower mRNA production. In addition, we find that the Pol II loading rate is significantly reduced in the mutant embryos compared to the wildtype (*Figure 4H*).

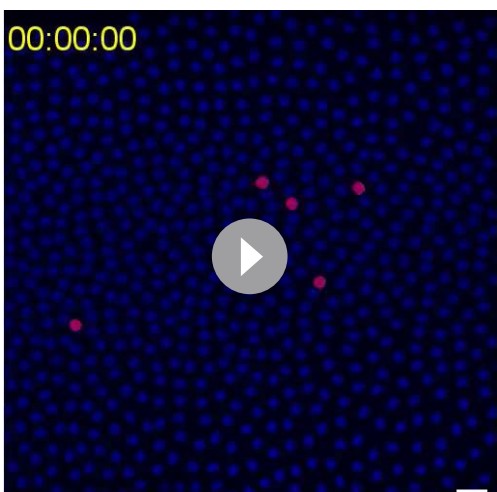

**Video 3.** False coloring of actively transcribing nuclei of a wildtype snamin> MS2-yellow embryo. Scale bar = 10 µm.

https://elifesciences.org/articles/85997/figures#video3

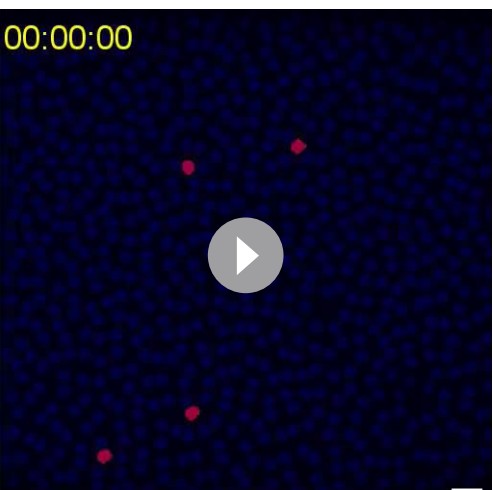

**Video 4.** False coloring of actively transcribing nuclei of a Dl2 mutant snamin> MS2-yellow embryo. Scale bar = 10 µm.

https://elifesciences.org/articles/85997/figures#video4

This confirms that the lower transcriptional amplitude mainly results from the promoter's inability to effectively load Pol II, along with an additional contribution from the reduced time the promoter spends in the ON state. In the full distal enhancer, we observe similar stochastic transcriptional activation, where the number of actively transcribing nuclei is about 50% less in the mutants than in the wildtype at any given time (*Figure 4—figure supplement 2A*). We also see a similar increase in the rate of $k_{off}$ and burst durations in the full distal enhancer, with minimal effect on $k_{on}$ (*Figure 4—figure supplement 2D–F*). The trend in the distributions of the promoter switching rates remains the same as well (*Figure 4—figure supplement 2B–C*). Taken together, our data demonstrate that TF-DNA binding modulates mRNA production by increasing the rate of promoter inactivation ($k_{off}$), along with the decrease in Pol II loading rate.

## Discussion

Organisms have evolved to contain enhancers with multiple binding sites for the same TF, not only for robustness under varying biological conditions, but as a molecular regulatory mechanism. Here, we studied transcriptional dynamics driven by different degrees of TF-DNA interactions through inducing mutations in the *sna* regulatory module. By utilizing quantitative live imaging, we dissected the effects of modulation of TF binding sites and determined that mRNA production of the target gene is drastically reduced when a single binding site is mutated. This reduction is mainly due to the decreased transcriptional amplitude, driven by a lower rate of Pol II loading, but is also slightly affected by shorter window of transcription, delayed transcriptional activation, and lower probability of activation (*Figure 2*, *Figure 4H*). Although a previous study demonstrated that an extended enhancer with additional binding sites could buffer against the effects of mutating a single Gt binding site (*López-Rivera et al., 2020*), we find that the same effect in the minimal enhancer is also observed in the full distal enhancer. This could be due to Dl and Twi's roles as activators, whereas the aforementioned study investigated Gt's role as a repressor. It is interesting to note that despite the presence of additional TF binding sites, modulating a single site can catastrophically reduce gene expression levels. We confirm that the minimal enhancer is the core region that is needed to recapitulate normal *sna* levels and that TFs in that region employ distinct mechanisms to regulate transcription in response to genetic perturbations.

Moreover, we determined that the TF binding sites in the *sna* enhancer work synergistically to drive the proper pattern and levels of the target gene expression. Cooperativity is a necessary mechanism by which a hub of weaker sites can coordinate to synergistically generate the correct expression pattern and levels in a developing embryo. In the context of *sna*, the *sna* enhancer only contains weak Dl binding sites, repressing *short gastrulation* (*sog*) expression in the region and preventing those cells from developing into neuroectoderm (*Hong et al., 2008*). Our thermodynamic model showed that increasing the binding affinity leads to ectopic expression (*Figure 3F*). Hence, it is crucial that the *sna* enhancer contains only weak Dl binding sites to drive expression exclusively in the region with high Dl concentration (i.e. mesoderm). Previous works have shown that other enhancers also utilize weak, sub-optimal binding sites to drive specific target gene expression. It was determined that despite the *shavenbaby* enhancers containing low-affinity binding sites, a microenvironment of high concentrations of Ubx and other cofactors can mediate efficient and specific transcription of the target locus (*Tsai et al., 2017*). Similarly, a single optimized Dl binding site in the *t48* enhancer resulted in earlier activity and ectopic expression patterns (*Keller et al., 2020*). Suboptimization of enhancers was shown to be an important characteristic of gene regulation to drive restricted expression, whereas optimizing TF motifs resulted in the loss of specificity and an increase in aberrant transcriptional activity (*Farley et al., 2015*).

Probing the thermodynamic model further reveals the importance of cooperativity in regulating expression. When all binding affinities in the *sna* enhancer are weakened, threefold higher cooperativity values are needed to maintain normal expression levels (*Table 2*). Similarly, in a *sna* enhancer with increased affinity binding sites, optimal cooperativity values to maintain normal expression levels and domains are ten-fold lower (*Figure 3—figure supplement 2A–B* and *Table 2*). Here, it is evident that concerted interaction among weaker affinity binding sites with high levels of cooperativity may be a mechanism to maintain a specific expression domain and level (*Figure 3D*). Such higher-degree of cooperativity among TFs has also been shown in a recent study, where cooperativity among Runt, Bcd, and Pol II are needed to drive normal gene expression level (*Kim et al., 2022*). Future experiments

involving binding site modulations of enhancers with stronger binding sites, such as those in the *sog* enhancer, can be performed to confirm the model results.

Interestingly, we found that the TF binding site mutations affect the rate of the promoter switching OFF. The heterogeneity in $k_{off}$ rates confirms that the mutations abolish the ability of the embryo to develop robustly by causing promoter instability and shorter transcription windows, leading to overall lower mRNA production. As a result of higher $k_{off}$ rates, the mutant embryos have fewer actively transcribing nuclei within the domain at any given time point (*Figure 4A and G Figure 4—figure supplement 1A*, *Figure 4—figure supplement 2A*, *Videos 3 and 4*). However, we note that the HMM only provides some explanation for the reduction in transcriptional activity since the changes in $k_{on}$ and $k_{off}$ are less drastic than the changes in transcriptional output. Since the amount of time the promoter spends in the ON state is not affected by the mutations, the lower transcriptional amplitude can be mainly attributed to the promoter's inability to effectively load Pol II (*Figure 2E*, *Figure 4D and H*). Furthermore, the shallower gradient of the expression pattern highlights the role of TFs in controlling the sharpness of the expression gradient, potentially affecting the robustness in germ layer formation (*Figure 1H and K*). In sum, we believe that multiple TF binding sites are imperative not only for pattern formation but also for eliminating extrinsic and intrinsic variabilities that may occur during development.

We illustrate that the minimal enhancer is a feasible model system to quantitatively and systematically study transcription kinetics and regulation. We note that although it may not always be possible to generalize the effects of mutations from a reporter construct to the endogenous setting, we can still gain valuable insights and broaden our understanding of transcription regulation. Since *sna* is responsible for mesoderm formation as well as for the repression of other patterning genes like *sog* that specify neuroectoderm, it will be interesting to characterize the phenotypic effects of these mutations. Specifically, if the mutation causes 65% reduction in *sna* activity endogenously, would the developing embryo undergo proper gastrulation and remain viable? However, since *sna* has at least two well-characterized enhancers (proximal and distal) that compensate for each other, investigating the phenotypic effects of endogenously modulating TF binding sites remains challenging. In order to correlate transcriptional dynamics with downstream development it will be critical to carefully design experiments that will disentangle the contributions of the individual enhancers and the role of specific TF binding sites.

In this work, we demonstrate that TFs can regulate transcriptional dynamics by tuning bursting parameters and modulating transcriptional activity in response to genetic perturbations. Using quantitative live imaging and thermodynamic modeling, we find that low-affinity TF binding sites can create an environment of increased transcriptional activity to drive localized, specific, and sharp expression patterns. The evidence of a dual modality of regulation and synergy highlights the importance of moving beyond fixed tissue studies and focusing on experiments that can tease apart subtle kinetic changes that occur during development. Collectively, our findings provide novel insights into enhancer-mediated transcriptional dynamics and expand our understanding of enhancer-TF binding through a combination of experimental and modeling approaches.

## Materials and methods

### Motif scanning

TF binding sites were found through the FIMO (Find Individual Motif Occurrences) (*Bailey et al., 2015*) tool using motifs from *Keller et al., 2020* and JASPAR (*Castro-Mondragon et al., 2022*). The cut-off p-value for motif match was set to p<1e-3. Mutations of the TF binding sites were confirmed by scanning the mutated sequence through FIMO and ensuring that it was no longer recognizable as a motif. The wildtype and mutated sequences are shown in *Table 1*.

### Plasmid and transgenic fly generation

The minimal distal *sna* enhancer was characterized in *Ferraro et al., 2016*. TF binding sites were mutated using PCR-mediated site-directed mutagenesis and confirmed via Sanger sequencing (*Table 1*). The mutated enhancers were cloned into a plasmid containing the core 100 bp *sna* promoter, 10 copies of MS2 stem loops, and the *yellow* reporter gene. Transgenic reporter lines were created

using PhiC31-mediated integration and the transgene was inserted to the VK33 locus (*Venken et al., 2006*). Injection was performed by the BestGene, Inc.

## Live imaging

Wild-type embryos were produced by crossing yw;His2Av-mRFP,nanos >MCP:GFP (*Fukaya et al., 2016*) virgin females to the desired y,w;MS2 males. The embryos from the cross were laid at 25 °C, dechorionated, and staged with Halocarbon oil. All images were taken using a Zeiss LSM800 confocal laser scanning microscope. Images were acquired with a Plan-Apochromat 40x1.3 NA oil objective using a 488 nm and 561 nm laser to visualize MCP:GFP and His2Av-mRFP, respectively, with a time resolution of 21 s/frame. Images were created using maximum projection of 14 z-stacks with 0.75 μm steps. The same exposure and laser settings were used for all minimal *sna* replicates and a different set of settings were used for all the full distal *sna* replicates. All images were acquired in 16-bits. Images were taken as the embryo entered the nuclear cycle 14 until the embryo began gastrulation.

## Quantification and statistical analysis

All the image processing methods and analyses were implemented in MATLAB (R2018b, Math-Works). Histograms of all the snapshots and movies shown in all figures were adjusted for visualization purposes only. Analyses of all data were performed using raw images. To determine statistical significance, the student's t-tests were performed. ** indicates p<0.001.

## Image analysis

Segmentation, nuclei tracking, and MS2 signal extraction were performed as described in *Syed et al., 2021*. Actively transcribing nuclei were labeled if they exceeded a predetermined fluorescence intensity threshold. mRNA production was calculated by integrating under the fluorescence intensity trajectories of actively transcribing nuclei. Activation time was defined as the time at which the MS2 signal increased beyond the threshold. Transcription window was calculated to be the time a nucleus was above the given threshold. The mean transcriptional amplitude was determined by averaging the MS2 signal for all transcriptionally active nuclei. The average transcriptional trajectory was obtained by averaging the intensity of all active nuclei at each timepoint. The sharpness of the expression gradient was determined by finding the maximum derivative of the mRNA production curves. The Pol II loading rate of an active nucleus was obtained by measuring the initial slope of the nucleus's smoothened fluorescence trajectory. The smoothened curve was interpolated by a factor of 10. The loading rate was determined to be the slope of the best-fit line after linear regression on the first 30 points above the threshold. Spatial analysis was performed by dividing the embryo into 16 bins along the dorsoventral axis. All the nuclei data within each bin was averaged to obtain the plots.

## Equilibrium binding model

The thermodynamic model utilized in this study is built on those described in *Eck et al., 2020*; *Kanodia et al., 2012*. Concentrations of nuclear Dl were assumed to follow a normal distribution. Twi concentration was calculated from *Lim et al., 2015* and normalized. The dissociation constants were chosen to reflect the relative affinity of each binding site based on their respective Patser scores (*Hertz and Stormo, 1999*). Assuming that the microstates are in equilibrium, the probability of transcription occurring is given by *Equation 1*:

$$p_{bound} = \frac{\frac{C_A}{K_A} + \frac{C_B}{K_B} + \frac{C_C}{K_C} + \frac{C_D}{K_D} + \omega_1\frac{C_AC_B}{K_AK_B} + \omega_2\frac{C_BC_C}{K_BK_C} + \omega_3\frac{C_AC_C}{K_AK_C} + \omega_4\frac{C_AC_D}{K_AK_D} + \omega_5\frac{C_BC_D}{K_BK_D} +}{1 + \frac{C_A}{K_A} + \frac{C_B}{K_B} + \frac{C_C}{K_C} + \frac{C_D}{K_D} + \omega_1\frac{C_AC_B}{K_AK_B} + \omega_2\frac{C_BC_C}{K_BK_C} + \omega_3\frac{C_AC_C}{K_AK_C} + \omega_4\frac{C_AC_D}{K_AK_D} + \omega_5\frac{C_BC_D}{K_BK_D}}$$

where $C_i$ is the concentration of a TF, $K_i$ is the dissociation constant, and $\omega_i$ is the cooperativity factor. To find the minimum of the nonlinear multivariable functions in the previous equation, we utilized a nonlinear programming solver, *fmincon* (MATLAB). The solver returns a vector of cooperativity values that minimize the objective function. The objective function uses the root mean square error between the wildtype and mutant conditions to determine the cooperativity terms that would satisfy the constraints (i.e., mutant condition must have 65% reduction in expression level compared to the wildtype). A stable solution was defined once the solution converged and the solver returned

cooperativities that satisfied the objective function and constraints within a step tolerance of 1e-10. Using these evaluated cooperativities, curves were generated to predict mRNA production. In the case of no cooperativity (*Figure 3B*), all cooperativity values were set to 1 and the results were plotted.

### Two-state model

Two-state model fitting is similar to that described in *Keller et al., 2020*. Transcriptional trajectories were smoothened using local regression (LOESS) method. Each trajectory from a given nucleus was converted into a binary plot indicating promoter ON (1) and promoter OFF (0) states as described below. The slope between two consecutive time-points of active transcription was obtained to define promoter ON and OFF states. Time points with positive slope were considered as ON promoter states (1), while those with a negative slope were assumed as OFF promoter states (0). This binary data was used as the input for the Baum-Welch based HMM. Initial transition probabilities were assumed to be equal (i.e., 0.5). These probabilities were adjusted in each iteration to individual burst traces until they converged. The Viterbi algorithm was used to determine the most likely sequence of (ON/OFF) promoter states (shown in *Figure 4C*). Burst separation and burst duration were correlated to $k_{on}$ and $k_{off}$, respectively.

## Acknowledgements

We thank members of the Lim lab for helpful discussions and comments on the manuscript. We especially thank Noel Buitrago for his help in imaging. Dl-venus fly was kindly provided by the Shvartsman Lab at Princeton University. We also thank the FlyBase for providing useful information (*Gramates et al., 2022*). This study was funded by NIH R35GM133425 awarded to SS and BL.

## Additional information

### Funding

| Funder | Grant reference number | Author |
| --- | --- | --- |
| National Institutes of Health | R35GM133425 | Sahla Syed<br>Bomyi Lim |

The funders had no role in study design, data collection and interpretation, or the decision to submit the work for publication.

### Author contributions

Sahla Syed, Conceptualization, Data curation, Formal analysis, Investigation, Visualization, Methodology, Writing - original draft, Writing - review and editing; Yifei Duan, Investigation, Visualization; Bomyi Lim, Conceptualization, Software, Supervision, Funding acquisition, Investigation, Methodology, Writing - original draft, Project administration, Writing - review and editing

### Author ORCIDs

Sahla Syed http://orcid.org/0000-0002-9917-0709
Bomyi Lim http://orcid.org/0000-0002-3058-9181

### Decision letter and Author response

Decision letter https://doi.org/10.7554/eLife.85997.sa1
Author response https://doi.org/10.7554/eLife.85997.sa2

## Additional files

### Supplementary files
• MDAR checklist

### Data availability
The scripts used in the paper are freely available on GitHub (copy archived at *Limlab-upenn, 2023*).

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
