## [Editor Report]

This valuable work explores how transcription factors regulate transcription through cooperative binding to enhancers. Through experiments and modeling, the authors show convincingly that the cooperativity of transcription factor binding regulates transcriptional bursting and the extent of the amount of time that the target promoter remains in an active state.

---

## [Decision Letter]

**Decision letter after peer review:**

Thank you for submitting your article "Modulation of protein-DNA binding reveals mechanisms of spatiotemporal gene control in early *Drosophila* embryos" for consideration by *eLife*. Your article has been reviewed by 2 peer reviewers, and the evaluation has been overseen by a Reviewing Editor and Michael Eisen as the Senior Editor. The reviewers have opted to remain anonymous.

Essential revisions:

1) The MWC model should be more extensively explored, including a more complete parameter sweep and better evaluation of what the values of the cooperativity parameters are.

2) The discussion needs to be expanded to include placing the work in a more general context: are the result relevant for other enhancers, in other organisms?

The reviewers make several suggestions to further strengthen the study.

*Reviewer #1 (Recommendations for the authors):*

In general, I thought this paper was nicely written with clear figures. I have a few comments to clarify some of the results and place the work in a larger context.

– I'm a little unclear about how direct the connection is between transcriptional amplitude and PolII loading rate. In theory, one could imagine that, as defined in this paper, the average MS2 signal could be higher in the WT enhancer either because the Pol II loading rate is higher or because the promoter is ON for a longer time, allowing the signal to accumulate. I think it's mostly the former, given the small changes in k_on_/k_off_, but it's hard to directly see in the data provided. Others have more directly looked at the initiation rate by measuring the positive slopes of the MS2 traces, which seems like it would be possible here and useful for a more clear insight into the Pol II loading rate.

– It would be more compelling to share a bit more of the results from the MWC model. It seems that there was a bit of an exploration of the parameter space, especially with regard to cooperativity, but the only thing shown is the final plots. How sensitive are the results to the parameters? How big are the cooperativity parameters? Do you have any sense of why the model predicts a bigger effect for the double binding site deletion, while the experimental results don't show this same effect? Is the impact the same for strong and weak binding sites?

– I would enjoy reading a discussion that explores just a bit further how general the observations from this work might be. Do you suspect that cooperativity might be more important in enhancers that need to have weak binding sites for specificity reasons (maybe somewhat testable from your MWC model)? Do enhancers that are sensitive to low concentrations of Dl show the same multiplicities of binding sites? Do you think the reason that the extended enhancer shows the same effects as the minimal enhancer has to do with Dl/Twi's roles as activators (in contrast to Gt's role as a repressor in eve stripe 2)?

*Reviewer #2 (Recommendations for the authors):*

– The equilibrium binding model is an important addition, yet is underdeveloped in the text and figures. The manuscript as written simply compares the "no cooperativity" condition with the fully solved "cooperativity" condition. What are the values of the cooperativity coefficients for the fitted results in Figure 3C? A sensitivity analysis would also be helpful to demonstrate the robustness of the results: e.g. how output changes by picking several intermediate cooperativity values. It would be nice to get an intuition of how promoter occupancy dynamics differ in the cooperative vs. non-cooperative scenarios. Out of curiosity (but not needed by any means), it might be interesting to compare promoter occupancy kinetics with the predicted bursting dynamics.

– Throughout the manuscript, the authors refer to "total mRNA production", however, this is an extrapolation from MS2 imaging data and not an actual measurement of mRNA molecules. It may bolster the manuscript to include smFISH data using probes against the yellow reporter gene to measure the actual number of mRNA produced.

– The authors delete many, but not all, TF motifs found in the minimal snail enhancer. Was there any rationale for the motifs chosen?

– Lines 125 and 382: manuscript reports using 10 MS2 repeats, Figure 1A says 24 repeats. The number of loops used should be clarified.

– In tracking the continuous MS2 signal, the authors report intensity values before that MS2 spot can be detected. How is this done? Are these intensity values derived from the background fluorescence seen in the nuclei?

---

## [Author Response]

Essential revisions:Reviewer #1 (Recommendations for the authors):In general, I thought this paper was nicely written with clear figures. I have a few comments to clarify some of the results and place the work in a larger context.– I'm a little unclear about how direct the connection is between transcriptional amplitude and PolII loading rate. In theory, one could imagine that, as defined in this paper, the average MS2 signal could be higher in the WT enhancer either because the Pol II loading rate is higher or because the promoter is ON for a longer time, allowing the signal to accumulate. I think it's mostly the former, given the small changes in k_on_/k_off_, but it's hard to directly see in the data provided. Others have more directly looked at the initiation rate by measuring the positive slopes of the MS2 traces, which seems like it would be possible here and useful for a more clear insight into the Pol II loading rate.

We thank the reviewer for their comments. As suggested by the reviewer, in order to connect transcriptional amplitude and Pol II loading rate, we measured the initiation rate by calculating the slope of the MS2 traces and correlated it to the Pol II loading rate. As expected, the initiation rate in wildtype is higher than in mutant embryos. This additional analysis suggests that the drastic reduction in transcriptional amplitude is due to the reduced Pol II loading rate, not k_on_, and corroborates the previously shown results and conclusions (Bothma et al., *PNAS* 2014, PMID: 24994903; Garcia et al., *Curr. Biol.* 2013, PMID: 24139738). We have added this plot in Figure 4H in the revised manuscript, which shows the initiation rates of the wildtype and mutant embryos, and revised the manuscript as follows.

We have added this in the Introduction (Page 4):

“We find that mutating a single TF (Dl or Twi) binding site in the enhancer significantly reduces mRNA production of the target gene, mainly through lowering transcriptional amplitude by reducing RNA polymerase (Pol) II loading rate, without significantly delaying the timing of initiation or affecting the probability of activation.”

We have added this in the Results (Page 14):

“Previously, we demonstrated that the mutations affect mRNA production through transcriptional amplitude (Figure 2E). This could be because either the mutations hinder the Pol II loading rate or reduce the time the promoter is in the ON state….

In addition, we find that the Pol II loading rate is significantly reduced in the mutant embryos compared to the wildtype (Figure 4H). This confirms that the lower transcriptional amplitude mainly results from the promoter’s inability to effectively load Pol II, along with an additional contribution from the reduced time the promoter spends in the ON state.”

We have added this in the Discussion (Page 15):

“This reduction is mainly due to the decreased transcriptional amplitude, driven by a lower rate of Pol II loading… and,

Since the amount of time the promoter spends in the ON state is not affected by the mutations, the lower transcriptional amplitude can be mainly attributed to the promoter’s inability to effectively load Pol II (Figure 2E, Figure 4D-F).”

– It would be more compelling to share a bit more of the results from the MWC model. It seems that there was a bit of an exploration of the parameter space, especially with regard to cooperativity, but the only thing shown is the final plots. How sensitive are the results to the parameters? How big are the cooperativity parameters? Do you have any sense of why the model predicts a bigger effect for the double binding site deletion, while the experimental results don't show this same effect? Is the impact the same for strong and weak binding sites?

We find that 3 cooperativity factors are larger than 1, indicating cooperativity. The cooperativity of Dl1/Dl2 (ω_1_) is 28, Dl3/Twi2 (ω_6_) is 55, and Dl1/Dl2/Dl3/Twi2 (ω_11_) is 1.6e5, suggesting that cooperativity of all binding sites is necessary to match experimental observations. Through a sensitivity analysis, we find that the model is robust to changes in parameters with moderate cooperativity values (ω_1_ and ω_6_), such that varying the cooperativity by an order of magnitude higher or lower did not result in significant changes. However, the model is somewhat sensitive to changes in the parameter with the highest cooperativity (ω_11_), the term representing interactions among all the TFs. Reduced cooperativity results in much smaller differences in expression levels between wildtype and mutants, whereas higher cooperativity causes ectopic expression in a wider expression domain (Figure 3—figure supplement 1).

We have added these results in the manuscript (Page 11):

“We found that Dl1/Dl2 (⍵1), Dl3/Twi2 (⍵6), and Dl1/Dl2/Dl3/Twi2 (⍵11) cooperativities were the key parameters with a value above 1 to reach a stable solution, confirming the role of cooperativity among weak binding sites as a mechanism for precise gene control (Table 2). We performed sensitivity analysis on these three cooperativity terms to test the robustness of the model. Transcriptional output was not significantly affected upon modulating a moderate cooperativity ⍵1 and ⍵6 values by an order of magnitude (Figure 3—figure supplement 1A, B). However, the model was more sensitive to changes in the highest cooperativity term ⍵11, which represents the interactions among all TFs (Table 2). Decreasing the cooperativity value of ⍵11 results in a smaller difference between wildtype and mutant transcriptional activities, while increasing cooperativity results in ectopic expression in a wider expression domain (Figure 3—figure supplement 1C). This underscores the importance of the concerted interaction among all the TFs in regulating and maintaining proper gene expression pattern and levels (Figure 3D).”

We do see a slight effect in the double binding site deletion (see Figure 2C-E and Figure 3C), but we think that this moderate difference is due to the fact that the enhancer has other low affinity binding sites that can buffer the predicted effect. The model only contains 4 binding sites, so removing two sites in the model would cause a more dramatic reduction than observed experimentally. We observe the same trend when all the binding sites are either strengthened or weakened (Figure 3—figure supplement 2A-B).

We note this difference in the revised manuscript (Page 10):

“We do note that the model predicts that the Dl1/2 double mutant exhibits lower mRNA output than the other single binding site mutants, which is not as prominent in experimental results (Figure 2C-E and Figure 3C). We believe this may be due to some degree of buffering from additional TF binding sites present in the sna enhancer that cannot be captured in our four binding site model. We observe the same trend of a lower mRNA output for the double binding site deletion regardless of the changes in binding site affinity (Figure 3—figure supplement 2A-B).”

– I would enjoy reading a discussion that explores just a bit further how general the observations from this work might be. Do you suspect that cooperativity might be more important in enhancers that need to have weak binding sites for specificity reasons (maybe somewhat testable from your MWC model)? Do enhancers that are sensitive to low concentrations of Dl show the same multiplicities of binding sites? Do you think the reason that the extended enhancer shows the same effects as the minimal enhancer has to do with Dl/Twi's roles as activators (in contrast to Gt's role as a repressor in eve stripe 2)?

While each binding site within the *snail* enhancer has varying affinities, they are all considered weak Dl and Twi binding sites compared to other known high affinity sites in other enhancers (i.e., the sequence doesn’t match the consensus site). Hence, we do not see significant differences in expression levels when removing a “weaker” Dl2 versus “stronger” Dl1 site experimentally (Figure 1G). To further explore the role of binding affinity strength on transcription, we utilized our model to strengthen or weaken one binding site and examined the changes in the predicted output. When we increase the binding affinity of one site (Dl1), removing the other weaker sites does not significantly affect the expression level. However, increasing Dl1 affinity results in a wider expression domain of *sna* and removal of that strong site (Dl1) results in a 70% reduction in transcription output (Figure 3—figure supplement 2F), an even larger reduction than the original experimental and model results. We hypothesize that the stronger site plays a more dominant role in controlling expression levels and driving ectopic expression. This may explain why the enhancer utilizes cooperativity among weaker sites rather than one strong site to regulate the target gene. Since Dl1 is already a weak site, weakening it even further causes it to effectively become almost a non-affinity site and leads to a reduction in overall mRNA production, and even further reductions upon binding site mutations (Figure 3—figure supplement 2E).

We have added the following in the Results (Page 12):

“Next, we utilized our model to test the effects of strengthening or weakening one binding site on transcriptional output. Further weakening Dl1 binding affinity renders it an almost non-affinity site and results in overall reduced mRNA production, while strengthening Dl1 affinity leads to an ectopically wider expression (Figure 3—figure supplement 2E-F). Relying on a single strong site may yield “too high” and ectopic expression, and its removal can drastically disrupt expression, a result that was also observed upon removal of a single Dl site in the t48 enhancer (Figure 3F) (Keller et al. 2020).”

The model predicts that cooperativity is indeed more important when the enhancer contains weak binding sites. We found that the cooperativity parameters needed to fit the experimental observations are about ten-fold higher when the enhancer contains binding sites that have all been weakened by a factor of 1.5 (Table 2). When binding affinity is increased by a factor of 2, we find that only ⍵_11_, the cooperativity term for interactions among all TFs, is significantly greater than 1, indicating that less cooperativity is needed for strong binding sites (Table 2). High gene expression level is observed only in the presence of strong cooperativity factors for enhancers with low affinity binding sites, emphasizing the need for cooperativity (Figure 3—figure supplement 2A and Table 2).

We have added the following in the Discussion (Page 16):

“Probing the thermodynamic model further reveals the importance of cooperativity in regulating expression. When all binding affinities in the sna enhancer are weakened, three-fold higher cooperativity values are needed to maintain normal expression levels (Table 2). Similarly, in a sna enhancer with increased affinity binding sites, optimal cooperativity values to maintain normal expression levels and domains are ten-fold lower (Figure 3—figure supplement 2A-B and Table 2). Here, it is evident that concerted interaction among weaker affinity binding sites with high levels of cooperativity may be a mechanism to maintain a specific expression domain and level (Figure 3D). Such higher degree of cooperativity among TFs has also been shown in a recent study, where cooperativity among Runt, Bcd, and Pol II are needed to drive normal gene expression level (Kim et al. 2022). Future experiments involving binding site modulations of enhancers with stronger binding sites, such as those in the sog enhancer, can be performed to confirm the model results.”

Interestingly, we find that the model is sensitive to Dl concentrations as the binding site affinity is decreased. In the presence of low Dl concentrations, enhancers with weak affinity binding sites will not interact with TFs as frequently, resulting in very low transcription (Figure 3—figure supplement 2C). Doubling Dl concentrations offsets the effects of low affinity sites and allows for normal expression levels. In the presence of high binding affinity sites, the effects of Dl concentration is less drastic, with only slightly wider expression domain upon increasing Dl concentrations (Figure 3—figure supplement 2D).

We have added these results in the Results (Page 11):

“We sought to obtain comprehensive insights into the mechanism of binding site interactions by systematically modulating model parameters. The sna enhancer contains all relatively weak Dl binding sites and activates expression only in the presence of high Dl concentration. To dissect the role of weak versus strong sites in enhancers regulating expression under different morphogen concentrations, we varied nuclear Dl levels. Decreasing Dl concentrations in the presence of low affinity binding sites results in a very low transcriptional activity, perhaps because the TF-DNA interactions are limited both by low TF concentration and low TF-DNA affinity (Figure 3—figure supplement 2C). In order to obtain high transcriptional activity under a low Dl concentration background, the affinity of the TF binding sites must be increased by a factor of 2 (Figure 3—figure supplement 2D). This suggests that enhancers of the genes activated by lower concentration of Dl must contain strong binding sites to maintain transcriptional activity at normal levels (Figure 3E) (Hong et al. 2008). It will be of further interest to quantitate how the induced mutations affect sna activity in the presence of reduced maternally deposited Dl levels.”

Lastly, we agree with the reviewer’s comment that the extended enhancer shows the same effects as the minimal enhancer, due to Dl/Twi’s roles as activators. These core 4 binding sites we tested seem to be sufficient to drive strong *sna* expression such that adding additional binding sites does not significantly alter the expression level.

We have included this in the Discussion (Page 15):

“This could be due to Dl and Twi’s roles as activators, whereas the aforementioned study investigated Gt’s role as a repressor.”

Reviewer #2 (Recommendations for the authors):– The equilibrium binding model is an important addition, yet is underdeveloped in the text and figures. The manuscript as written simply compares the "no cooperativity" condition with the fully solved "cooperativity" condition. What are the values of the cooperativity coefficients for the fitted results in Figure 3C? A sensitivity analysis would also be helpful to demonstrate the robustness of the results: e.g. how output changes by picking several intermediate cooperativity values.

We thank the reviewer for helpful suggestions. We find that 3 cooperativity factors are larger than 1, indicating cooperativity. The cooperativity of Dl1/Dl2 (ω_1_) is 28, Dl3/Twi2 (ω_6_) is 55, and Dl1/Dl2/Dl3/Twi2 (ω_11_) is 1.6e5, suggesting that cooperativity of all binding sites is necessary to match experimental observations. Through a sensitivity analysis, we find that the model is robust to changes in parameters with moderate cooperativity values (ω_1_ and ω_6_), such that varying the cooperativity by an order of magnitude higher or lower did not result in significant changes. However, the model is somewhat sensitive to changes in the parameter with the highest cooperativity (ω_11_), the term representing interactions among all the TFs. Reduced cooperativity results in much smaller differences in expression levels between wildtype and mutants, whereas higher cooperativity causes ectopic expression in a wider expression domain (Figure 3—figure supplement 1).

We have added these results in the manuscript to reflect our findings (Page 11):

“We found that Dl1/Dl2 (⍵1), Dl3/Twi2 (⍵6), and Dl1/Dl2/Dl3/Twi2 (⍵11) cooperativities were the key parameters with a value above 1 to reach a stable solution, confirming the role of cooperativity among weak binding sites as a mechanism for precise gene control (Table 2). We performed sensitivity analysis on these three cooperativity terms to test the robustness of the model. Transcriptional output was not significantly affected upon modulating a moderate cooperativity ⍵1 and ⍵6 values by an order of magnitude (Figure 3—figure supplement 1A, B). However, the model was more sensitive to changes in the highest cooperativity term ⍵11, which represents the interactions among all TFs (Table 2). Decreasing the cooperativity value of ⍵11 results in a smaller difference between wildtype and mutant transcriptional activities, while increasing cooperativity results in ectopic expression in a wider expression domain (Figure 3—figure supplement 1C). This underscores the importance of the concerted interaction among all the TFs in regulating and maintaining proper gene expression pattern and levels (Figure 3D).”

We have added the following to the main text (Page 10):

“Promoter occupancy is significantly higher when TFs interact cooperatively, such that removing cooperativity reduces promoter occupancy by more than 90% (Table 3).”

It would be nice to get an intuition of how promoter occupancy dynamics differ in the cooperative vs. non-cooperative scenarios. Out of curiosity (but not needed by any means), it might be interesting to compare promoter occupancy kinetics with the predicted bursting dynamics.

In cooperative scenarios, the promoter occupancy is significantly increased by the cooperativity term, whereas in non-cooperative scenarios, the occupancy is reduced by more than 90%. Since only 3 cooperativity terms are larger than 1, we only performed this analysis on those 3.

We have included Table 3 with the values of those states with and without cooperativity. As expected, since the cooperativity among all TFs (□_11_) is very large, the promoter occupancy in the non-cooperative state is very low.

– Throughout the manuscript, the authors refer to "total mRNA production", however, this is an extrapolation from MS2 imaging data and not an actual measurement of mRNA molecules. It may bolster the manuscript to include smFISH data using probes against the yellow reporter gene to measure the actual number of mRNA produced.

We thank the reviewer for the suggestion. Previous studies have correlated smFISH with MS2 live imaging data to obtain an absolute number of mRNA molecules (Garcia et al., *Curr. Biol.* 2013, PMID: 24139738; Bothma et al., *PNAS* 2014, PMID: 24994903). However, they report large errors in their estimations of mRNA production. We understand that utilizing the MS2 fluorescence intensity curves does not provide an exact number of mRNAs. However, we quantified the relative difference in RNA production among different mutant and wildtype embryos under the same microscope laser setting. Hence, we do not believe that an absolute number is necessary to illustrate the main points of the paper.

We have updated the text and figures to reflect this and have replaced “total mRNA production” with “mRNA production.” We also included more detail on correlating mRNA production with the MS2 fluorescence trajectories in the Introduction (Page 6).

Integration of the transcription trajectory over a given duration is proportional to the total number of mRNA molecules produced by a single nucleus (Bothma et al. 2014; Garcia et al. 2013). We do note that this does not provide an absolute number of mRNA molecules. However, through this estimation of mRNA production measured under the same laser setting, we can compare the relative mRNA production among constructs with different TF binding site mutations.

– The authors delete many, but not all, TF motifs found in the minimal snail enhancer. Was there any rationale for the motifs chosen?

These motifs were chosen based on their binding affinities. We studied the sites that had higher affinities than other TF binding sites found within the minimal enhancer. This is reflected in the main text (Page 5):

“The minimal enhancer contains binding sites for many TFs, including Dl, Twi, and Zld, and our study focuses on the sites with the strongest binding affinities, three Dl sites and one Twi site (Figure 1A).”

– Lines 125 and 382: manuscript reports using 10 MS2 repeats, Figure 1A says 24 repeats. The number of loops used should be clarified.

We thank the reviewer for bringing this to our attention and it has been clarified. We used 10 MS2 repeats, have corrected Figure 1A.

– In tracking the continuous MS2 signal, the authors report intensity values before that MS2 spot can be detected. How is this done? Are these intensity values derived from the background fluorescence seen in the nuclei?

The intensity values that are reported prior to the MS2 visualization are due to background fluorescence (diffused MCP signals). Even if we subtracted the background value, non-zero intensity values are still measured due to noise. In Methods, we have cited a previous paper that details how MS2 fluorescence was extracted from individual nuclei.